# A Novel Photoelectrochemical Aptamer Sensor Based on CdTe Quantum Dots Enhancement and Exonuclease I-Assisted Signal Amplification for *Listeria monocytogenes* Detection

**DOI:** 10.3390/foods10122896

**Published:** 2021-11-23

**Authors:** Liangliang Zhu, Hongshun Hao, Chao Ding, Hanwei Gan, Shuting Jiang, Gongliang Zhang, Jingran Bi, Shuang Yan, Hongman Hou

**Affiliations:** 1Department of Inorganic Nonmetallic Materials Engineering, Dalian Polytechnic University, Dalian 116034, China; zhull72@163.com (L.Z.); dc1608658525@163.com (C.D.); ghw1717@163.com (H.G.); JJiangSHuTing@163.com (S.J.); yanye150@outlook.com (S.Y.); 2Liaoning Key Lab for Aquatic Processing Quality and Safety, School of Food Science and Technology, Dalian Polytechnic University, Dalian 116034, China; zhanggl1978@hotmail.com (G.Z.); bijingran1225@foxmail.com (J.B.); houhongman@dlpu.edu.cn (H.H.)

**Keywords:** photoelectrochemical detection, aptamer, Listeria monocytogenes, quantum dots, exonuclease I

## Abstract

To achieve the rapid detection of *Listeria monocytogenes*, this study used aptamers for the original identification and built a photoelectrochemical aptamer sensor using exonuclease-assisted amplification. Tungsten trioxide (WO_3_) was used as a photosensitive material, was modified with gold nanoparticles to immobilize complementary DNA, and amplified the signal by means of the sensitization effect of CdTe quantum dots and the shearing effect of Exonuclease I (Exo I) to achieve high-sensitivity detection. This strategy had a detection limit of 45 CFU/mL in the concentration range of 1.3 × 10^1^–1.3 × 10^7^ CFU/mL. The construction strategy provides a new way to detect *Listeria monocytogenes*.

## 1. Introduction

Foodborne illness induced by pollution is a serious safety issue [1]. Pathogens in food can cause the food to spoil and rot, and some can produce toxic substances that can lead to illness. In recent years, food-borne poisoning incidents due to *Listeria monocytogenes* have aroused widespread concern [2]. *Listeria monocytogenes* can survive and grow under both aerobic and anaerobic conditions. It grows in a wide temperature range and has strong resistance to alkali and salt [3]. It is widely found in milk; meat; aquatic products; and frozen, and cold storage, and ready-to-eat foods [4]. *Listeria monocytogenes* can cause meningitis and sepsis, which have high hospitalization rates and mortality [5,6,7]. Various countries have formulated relevant limit standards. In China, it is stipulated that *Listeria monocytogenes* cannot be detected in meat products (GB29921-2013), as is true in the United States [8].

Traditional microbial detection methods include separation, culture, detection, and other steps. Although the detection accuracy is high, the detection time is long and the subsequent detection steps are cumbersome, which is not conducive to rapid detection [9]. In recent years, many rapid detection methods have been developed for pathogenic bacteria. These are mainly immunological detection methods based on the enzyme-linked immunosorbent method and the enzyme-linked fluorescence analysis method [10], along with molecular biological detection based on PCR technology and loop-mediated isothermal amplification technology [9,11,12,13]. However, all these methods need special reagents, instruments, and equipment, which are operated by professionals. There is a very important, practical need to explore more convenient, sensitive, effective, and rapid detection methods. Because of their good sensitivity, rapidity, and easy operation, biosensors have been developed and widely used. They have been used in medicine, environmental testing, food safety, and other fields [14,15,16,17,18,19]. The photoelectrochemical (PEC) sensor is a new type of detection technology with development potential. Because of the separation of the excitation light source and the generated photocurrent, the PEC sensor has a higher sensitivity than the electrochemical sensor and has great development potential [20]. PEC sensors have been used to detect DNA [21,22], miRNA [23,24], small organic molecules [25,26], heavy metal ions [27,28,29], proteins [30,31], and other substances, and they have attracted much attention.

Choosing optoelectronic materials with good photoelectric activity is one of the keys to improving sensor performance [32]. Graphite phase carbon nitride [33,34,35], bismuth-based semiconductors [36,37], and sulfides [30,38] are all considered potential materials. WO_3_ is a typical n-type tungsten oxide semiconductor that has good chemical stability, strong electron transmission ability, and a proper band gap (2.5–2.8 eV). It has been widely used in photocatalysis and sensors, as well as in other fields. Shen et al. [39] constructed a WO_3_-doped gold nanoparticle gas sensor that can detect ppb-level NO_2_. Lu et al. [40] prepared a core-shell WO_3_/CdS heterojunction, which also had excellent photocatalytic performance in the near-infrared region. Gold nanoparticles (Au NPs) have good electrical conductivity and the ability to fix identification elements. Therefore, they are often used to construct sensors and are promising intermediates [41].

Because of their characteristics of the narrow band gap, high yield, wide absorption, and adjustable size, quantum dots (QDs) have been widely used in the field of sensors [42,43]. Quantum dots are a kind of nanoscale semiconductor, such as CdS, CdSe and CdTe. In addition, QDs can be labeled with various biomolecules after chemical modification, and their good biocompatibility has unique advantages in biosensor applications. Aptamers are low cost, have high selectivity and good specificity, and are excellent substitutes for antibodies. In addition, aptamers have a variety of targets, which can be used to identify proteins, amino acids, pathogenic bacteria, cells, and viruses [44]. Exonuclease I (Exo I) can degrade single-stranded DNA in the 3′→5′ direction, and it only recognizes the 3′ end of single-stranded DNA. Compared with other cutting enzymes, its operation is simple, so it is also widely used [45].

In this study, a PEC aptamer sensor based on WO_3_, CdTe QDs, and Exo I auxiliary signal amplification was constructed to detect *Listeria monocytogenes*. WO_3_ was used as a photosensitive material and modified with Au NPs as intermediates for connecting the complementary DNA (cDNA) of the aptamer. Then, the quantum dot-coupled aptamer (QD–Ap) is modified on the electrode by DNA hybridization. The photocurrent is significantly enhanced because of the sensitization of quantum dots. When the target bacteria and Exo I are present, the photocurrent is significantly reduced. The constructed sensing platform provides a new strategy with a high potential for the rapid and efficient detection of *Listeria monocytogenes* in food.

## 2. Experimental Section

### 2.1. Chemicals and Reagents

Sodium tungstate dihydrate (Na_2_WO_4_·2H_2_O), ammonium oxalate monohydrate ((NH_4_)_2_C_2_O_4_·H_2_O), ascorbic acid (AA), cadmium chloride hemi(pentahydrate) (CdCl_2_·2.5H_2_O), and 6-mercapto-1-hexanol (MCH) were obtained from Macklin (Shanghai, China). Potassium tellurite (K_2_TeO_3_), chloroauric acid (HAuCl_4_), 1-ethyl-3(3-dimethylaminopropyl) carbodiimide hydrochloride (EDC), ethanol, Tris (2-carboxyethyl) phosphine (TCEP), and *N*-hydroxysuccinimide (NHS) were purchased from Aladdin (Shanghai, China). Sodium borohydride (NaBH_4_) and thioglycolic acid (TGA) were obtained from Sinopharm Chemical Reagent Co., Ltd. (Shanghai, China). Tryptone, yeast extract powder, and agar (bacteriological) were obtained from Qingdao Hope Bio-Technology Co., Ltd. (Qingdao, China). Tris HCl solution, Exonuclease I (Exo I), and phosphate-buffered saline (PBS) were purchased from Sangon Biotech Co., Ltd. (Shanghai, China). Raw chicken was purchased from a local supermarket. *Listeria monocytogenes* (ATCC 19115), *Staphylococcus aureus*, *Escherichia coli O157:H7*, and *Salmonella typhimurium* were obtained from China General Microbiological Culture Collection Center (Beijing, China). The DNA strands used were ordered from Sangon Biotech Co., Ltd. (Shanghai, China). All DNA sequences are summarized as follows: aptamer (Ap), 5′-NH_2_-(CH_2_)_6_-ATA CCA GCT TAT TCA ATT CCA AAA GCG CAC CCA TAT ATG TTC TAT GTC CCC CAC CTC GAG ATT GCA CTT ACT ATC T-3′ [46]; and aptamer-complementary DNA (cDNA), 5′-GTG CGC TTT TGG AAT TGA ATA AGC TGG TAT TTT TTT-(CH_2_)_6_-SH-3′.

### 2.2. Apparatus

The X-ray diffraction (XRD) patterns of samples were measured using a 7000S device (Shimadzu, Japan). The test conditions were set with a Cu-Ka XRD source, a scan rate of 5°/min, and a scan range of 10–70 degrees at room temperature. Surface morphologies of the prepared materials were observed using field emission scanning electron microscopy (JSM-7800F, Tokyo, Japan). Transmission electron microscopy (TEM) was performed using a JEOL JEM-2100 UHR (Tokyo, Japan). The UV–visible (UV–vis) absorption spectra were recorded using a TU-1901 UV–visible spectrophotometer (Beijing, China). Photoluminescence (PL) spectroscopy was characterized by an F-7000 fluorescence spectrometer (Hitachi, Japan) at room temperature. A xenon lamp was applied as the excitation source for the photoelectrochemical tests. Photocurrent and electrochemical characterizations were conducted with a CHI 660C electrochemical workstation (Shanghai, China) with a typical three-electrode system.

### 2.3. Preparation of WO_3_ Nanoplate

First, 0.4 g of Na_2_WO_4_·2H_2_O and 0.17 g of (NH_4_)_2_C_2_O_4_·H_2_O was dissolved in 33 mL deionized water. After stirring for 10 min, 9 mL of HCl solution (37%) was added. After stirring for another 10 min, 8 mL of H_2_O_2_ (30%) was added. Stirring was continued for 20 min. Then, 30 mL of absolute ethanol was added, and the solution was stirred for 30 min. The conductive surface of the pretreated SnO_2_ transparent conductive glass doped with fluorine conductive glass (FTO) was facing downward, inclined at 45° close to the beaker wall, placed in the above solution. This was placed in a constant temperature bath at 85 °C for 200 min and was then washed with deionized water and dried at 60 °C. Finally, the WO_3_/FTO electrode was prepared by calcining at 500 °C for two hours in a muffle furnace, cooling to room temperature, washing with deionized water, and drying.

### 2.4. Preparation of Au NPs

Two milliliters of 50 mmol/L HAuCl_4_ solutions were added into a double-necked flask with 98 mL deionized water. Stirring and refluxing were performed in an oil bath at 120 °C. When reflux began, the sodium citrate solution was added (10 mL, 38.8 Mm). When the solution turned wine-red, after refluxing for 20 min, the solution was cooled to room temperature and stored in a refrigerator.

### 2.5. Preparation of CdTe QDs

CdCl_2_·2.5H_2_O (0.0457 g) was dissolved in 50 mL of deionized water; then, 18 µL of TGA (thioglycolic acid) was added, and the solution was stirred for 10 min. The pH was then adjusted to 10.7 with 1 M NaOH. K_2_TeO_3_ (0.010 g) was dissolved in deionized water (50 mL) and added to the above solution after fully stirring. After 5 min, 80 mg NaBH_4_ was added. Then, the flask was connected to the condenser and condensed at 100 °C for 3 h. After cooling to room temperature, the TGA-coated CdTe QDs were centrifugally washed with ethanol. Then, the same amount of deionized water was added, and the solution was stored in a refrigerator at 4 °C.

### 2.6. Preparation of the CdTe QD–Ap Conjugates

Quantum dots (QDs) were coupled with aptamers (Ap) through an EDC–NHS coupling reaction. CdTe QDs (400 μL) were activated by 30 μL of 40 mM EDC and 30 μL of 15 mM NHS for 1 h. Then, 40 μL of 2 μM Ap was added, and the solution was placed in 4 °C shakers for 12 h. Finally, the solution was purified to obtain the QD–Ap conjugate.

### 2.7. The Culture Process of Listeria Monocytogenes

First, 1 g tryptone, 0.5 g yeast extract powder, and 1 g NaCl were added to 100 mL of deionized water, and the pH was adjusted to 7.5 with NaOH. After sterilization, the configuration of the LB liquid medium was completed. The solid medium was prepared with 1 g tryptone, 0.5 g yeast extract powder, 1 g NaCl, 1.5 g agar powder, and 100 mL deionized water. The pH was adjusted to 7.5, and the medium was sterilized with high-pressure steam. The purchased *Listeria monocytogenes* lyophilized powder was stored with glycerin. Next, 20% glycerin was added to the bacterial solution, and the mixture was evenly mixed, divided into cryogenic tubes, and stored in a refrigerator at −80 °C. Before the experiment, an ultra-clean worktable was sterilized by ultraviolet irradiation for 30 min, and the *Listeria monocytogenes* frozen solution was thawed naturally at room temperature. After sterilization, 100 μL of thawing solution was added to the liquid medium, and then it was shaken in a shaking table at 37 °C for 15 h activation. The configured solid medium was poured into a Petri dish plate next to an alcohol lamp on the ultra-clean workbench to allow it to solidify naturally. After the plate was completely solidified, the secondary activation solution was picked up with the inoculation ring and streaked on the plate. Then, the plate was placed in a 37 °C incubator for 12 h. A single *Listeria monocytogenes* colony was added into the liquid medium, which was shaken on a shaking table at 37 °C for 15 h. Thus, the *Listeria monocytogenes* stock solution culture was completed. A series of 10 times gradient dilutions of *Listeria monocytogenes* stock solution was carried out.

### 2.8. Fabrication of the Aptamer Sensor

The working electrode was constructed as illustrated in Figure 1. First, 30 µL of Au NPs was dropped onto the surface of the WO_3_/FTO electrode. Then, 30 µL of cDNA was mixed with TCEP (0.6 µL, 10 mM) and dropped onto the surface of the Au/WO_3_/FTO electrode and incubated. After that, the electrode was washed with Tris HCl to remove the unconnected cDNA. Next, 30 µL of 1 mM 6-mercapto-1-hexanol (MCH) was added dropwise to the electrode, sealed for 1 h, and washed with Tris HCl to obtain an MCH/cDNA/Au/WO_3_/FTO electrode. Next, 30 µL of CdTe Quantum dot–aptamers (QD–Aps) was added to the MCH/cDNA/Au/WO_3_/FTO electrode and incubated at 4 °C for 1 h to hybridize the cDNA with Ap. When the QD–Ap/MCH/cDNA/Au/WO_3_/FTO electrode was obtained, photocurrent detection was carried out, and the detection value was recorded as a. After rinsing with Tris HCl, 30 µL of pathogenic bacteria solutions of different concentrations containing 20 U of Exonuclease I (Exo I) was dropped onto the QD–AP/MCH/cDNA/Au/WO_3_/FTO electrode and incubated for 60 min. After rinsing with Tris HCl, the electrode was placed in a refrigerator at 4 °C for photocurrent detection in the next step, and the photocurrent was recorded as b. The current change, detected twice, was recorded as ΔI = a–b. Finally, the relationship between ΔI and the concentration of *Listeria monocytogenes* was calculated and analyzed.

### 2.9. Recovery Experiments

A 25 g chicken sample was placed in a sterile homogenization bag containing 225 mL of LB broth without additives and homogenized continuously on a flapping homogenizer for 2 min, without incubation of the chicken homogenate after contamination. A quantity of 1 mL of 1.3 × 10^1^ CFU/mL, 1.3 × 10^4^ CFU/mL, or 1.3 × 10^7^ CFU/mL *Listeria monocytogenes* liquid was added to 9 mL of sample homogenate, and the test tube was shaken to mix it evenly. The sample homogenate containing different concentrations of pathogenic bacteria was centrifuged for 5 min under low-speed centrifugation at 2000 r/min. The supernatant was absorbed and added into the sterilization centrifuge tube and centrifuged at a speed of 12,000 r/min for 3 min. After that, the supernatant was discarded, and 1 mL of normal saline was added for full mixing. Then the *Listeria monocytogenes* pollution solution of the adult model was configured, and the prepared sensor was used for analysis and detection.

## 3. Results and Discussion

### 3.1. Photoelectrochemical Reaction Mechanism of the Sensor

The charge transfer of the working electrode is displayed in Figure 2. WO_3_ has a high electron-hole recombination rate and a low photoelectric conversion efficiency. Therefore, sensitizing the CdTe quantum dots with a small band gap can broaden the spectral response range, improve the photoelectric conversion efficiency, and increase the photocurrent intensity. Au NPs mainly connect aptamers through Au-S bonds and can also increase the absorption of light by quantum dots. AA is used as the electron donor in the electrolyte solution.

Under illumination, the electrons generated by the CdTe quantum dots’ transition from the valence band to the conduction band flow through the Au NPs and are injected into the conduction band of WO_3_. Then, together with the photogenerated electrons generated inside WO_3_, they are finally transferred to the FTO electrode to generate the current signal. The WO_3_ and CdTe valence bands leave a large number of photogenerated holes. At this time, the AA in the solution provides electrons and is oxidized by the holes.

When there were no *Listeria monocytogenes*, the CdTe QDs were close to the electrode surface and produced sensitization, and the photocurrent increased. In the presence of *Listeria monocytogenes*, Ap specifically combined with *Listeria monocytogenes*, causing the CdTe QDs to leave the electrode surface and reduce the photocurrent intensity. Meanwhile, Exo I could recognize and cut Ap. *Listeria monocytogenes* bound to Ap continued to bind to Ap on the electrode, and the QDs were far away from the electrode. In this cycle, the QDs on the electrode gradually decreased, and the photocurrent decreased significantly. The sensitive detection of *Listeria monocytogenes* was achieved.

### 3.2. Characterization of Synthesized Materials

Figure 3a shows the XRD patterns of FTO, WO_3_/FTO, and Au/WO_3_/FTO. The WO_3_/FTO diffraction peaks (002), (200), (020), (-112), (202), (222), (140), (240), and (420) are obvious in the figure, which proves the formation of monoclinic crystal WO_3_ (JCPDS no. 43-1035). When the Au NPs were modified, no new peaks appeared, but the intensity of the WO_3_ peaks was significantly reduced. This may be because of the Au NP layer covering the surface of WO_3_, but the content of Au NPs was too small. Figure 3b shows the XRD pattern of CdTe QDs. The 2θ value corresponds to the three crystal planes (111), (220) and (311) of the cubic crystal CdTe on the standard card no. 65-1046. The above results indicate that the sensor electrode material was successfully prepared.

Figure 4 shows the SEM images of WO_3_ and Au/WO_3_ modified on FTO conductive glass. It can be seen that WO_3_ nanosheets were arranged vertically on the surface of the FTO glass (Figure 4a) with regular shapes and a thickness of 70.8–74.7 nm. Figure 4b shows a cross-sectional SEM image of WO_3_/FTO. The WO_3_ layer deposited on the surface of FTO was relatively uniform, with a thickness of about 422 nm. The vertically arranged structure increased the surface area of the WO_3_ layer and could improve the utilization rate of light. After modification with Au NPs, as shown in Figure 4c, the basic morphology of the WO_3_ layer did not change, and the surface of the smooth nanosheet became rough, indicating that the Au NPs were successfully modified. There was no change in the cross-sectional thickness after the modification of Au NPs.

The TEM image (Figure 5a) of Au NPs shows that the prepared Au NPs were spherical and uniform in size, and the particle size was between 10 and 15 nm. Obvious lattice fringes can be seen in Figure 4b, and the lattice spacing of the Au (111) lattice plane was about 0.235 nm after software analysis.

To further confirm the surface element composition, we performed an XPS test on WO_3_/Au. Figure 6a shows the full spectrum of WO_3_/Au. W, Au, O, and C were observed. Figure 6b shows the XPS spectrum of W 4f. It shows that the characteristic peaks of the binding energy of W 4f_7/2_ and W 4f_5/2_ were 35.7 eV and 37.7 eV, consistent with W^6+^. The XPS spectrum (Figure 6c) of O 1s showed two characteristic peaks of the binding energies of 530.3 eV and 531.9 eV. The peak at 530.3 eV aligns with the O^2−^ ions characteristic of the WO_3_ phase, corresponding to lattice oxygen. The peak at 531.9 eV corresponds with the chemically adsorbed oxygen species at the oxygen vacancy in WO_3_ [41,47,48]. The XPS spectrum of Au 4f consisted of two parts, as shown in Figure 6d. The characteristic peaks at 83.8 eV and 87.5 eV can be ascribed to Au 4f_7/2_ and Au 4f_5/2_, respectively.

Figure 7 shows the UV–vis (a) and fluorescence spectra (b) of CdTe QDs. The curves a, b and c in the figure represent quantum dots prepared through reactions for 30 min, 1 h and 3 h, respectively. With the extension of the reaction time, the peaks of the ultraviolet–visible absorption spectrum and the fluorescence spectrum gradually shifted to the long-wavelength direction, which enhanced the absorption range of visible light. This may be due to the quantum confinement effect, through which the emission wavelength was red-shifted. The stronger the peak in the fluorescence spectrum, the higher the electron-hole recombination rate. After optimization experiments, CdTe QDs with a reflow time of 3 h were selected to construct the sensor. The exciting absorption peak was 538 nm, and its maximum fluorescence emission wavelength was 582 nm.

### 3.3. Photocurrent Characterization of the Aptamer Sensor

In order to prove the successful preparation of the working electrode of the aptamer sensor, a photocurrent response test was performed in a 10 × PBS buffer containing 0.12 mol/L of AA with a pH of 7.4. As shown in Figure 8, the light source was turned on and off at 20 s and 40 s, and photocurrent changes within 60 s were observed. The blank FTO had no change in photocurrent (curve a), and when modified with WO_3_, the photocurrent significantly increased (curve b). This is because WO_3_ had good photoelectric activity and generated current under light conditions. When Au NPs were modified on the electrode surface, the photocurrent decreased (curve c). Since the Fermi level of gold nanoparticles is lower than that of WO_3_, the work function of gold nanoparticles is greater than that of WO_3_. To balance the two Fermis levels, part of the photogenerated electrons on the conduction band of WO_3_ is transferred to the gold nanoparticles, which causes a decrease in the photocurrent. When cDNA and MCH were modified, the photocurrent had a slight increase (curve d), which may have been due to the modification of cDNA and MCH, weakening the balance trend of the Fermi level [49]. Therefore, the tendency of photogenerated electrons on the conduction band of WO_3_ to transfer to gold nanoparticles was weakened. After the QD–Ap conjugate was modified on the electrode, the photocurrent was significantly enhanced because of the sensitization of the quantum dots (curve e). When 30 μL of 1.3 × 10^6^ CFU/mL *Listeria monocytogenes* (LM) solution containing 20 U of Exo I was added dropwise and incubated for 60 min, the photocurrent decreased significantly (curve f). This is because the specific binding of *Listeria monocytogenes* and Ap caused the QD–Ap conjugate to detach from the electrode surface, and the quantum dot sensitization was weakened. The shearing effect of Exo I also released the *Listeria monocytogenes* that had been bound to Ap, which re-attached to the electrode surface. The binding of Ap further weakened the sensitization effect. To verify the shear cycle amplification effect of Exo I, a control experiment was carried out. As shown in Figure 9, the photocurrent of the aptamer electrode (curve b) with 30 µL of 1.3 × 10^6^ CFU/mL *Listeria monocytogenes* containing Exo I was 26% lower than that of the electrode without Exo I (curve a). This shows that Exo I has a significant signal-amplifying effect on the photocurrent detection process. In summary, this shows that the aptamer sensor was successfully constructed and can be used for *Listeria monocytogenes* detection.

### 3.4. EIS and CV Characterization of Aptamer Sensors

Electrochemical impedance spectroscopy (EIS) was used to further prove the successful construction of the aptamer sensor. EIS was used to a different frequency AC signal to the system; analyze the change in impedance with frequency; analyze the electrode dynamics, diffusion, and electric double layer; and study the mechanism of the solid electrolyte and corrosion protection electrode materials. Electrochemical impedance analysis was performed on working electrodes with different modification processes in solutions. A typical EIS spectrum is a curve with a semicircle and a “tail”, which correspond to the high-frequency region and the low-frequency region, respectively. The high-frequency area is dominated by charge transfer, and the low-frequency area is dominated by mass transfer. Among them, the diameter of the high-frequency region circle is equal to the charge transfer resistance (Rct). The larger the diameter, the greater the obstruction of the oxidation–reduction probe on the electrode surface. In Figure 10, FTO has the smallest diameter (curve a), indicating that FTO without any modification can transfer electrons more effectively. After the deposition of WO_3_, the impedance increased (curve b). Because of the excellent conductivity of gold nanoparticles, the impedance decreased after modification with Au NPs (curve c). The impedance value increased after cDNA and MCH were modified (curve d). This is because the oligonucleotide was negatively charged, and the oxidation–reduction probe Fe (CN)_6_^3−/4−^ was also negatively charged; the repulsive force between the two caused an increase in the resistance of electron transport, and MCH is non-conductive, which led to an increase in impedance. When the QD–Ap conjugate was modified on the electrode, the impedance was further increased because of the weak conductivity of QDs and the increase in oligonucleotides. The impedance decreased after incubating Exo I and *Listeria monocytogenes*. This is because many oligonucleotides and QDs left the electrode surface after the specific binding of *Listeria monocytogenes* and Ap and Exo I shearing. Therefore, the change of impedance value proves that the aptamer sensor was constructed successfully.

Cyclic voltammetry is also a commonly used electrochemical analysis method, as shown in Figure 11. On the bare FTO electrode (curve a), a pair of obvious redox anode and cathode peaks can be observed at −0.15 V and 0.5 V, and the two peaks are symmetrical, which proves that the reaction is reversible. After being modified with WO_3_, the redox peak current decreased (curve b). When Au NPs were modified, the current value increased (curve c), indicating that Au NPs can accelerate electron transfer. When cDNA, MCH (curve d), and QD–Ap conjugate (curve e) were modified sequentially, the redox peak current gradually decreased, and the current value increased after incubating Exo I and *Listeria monocytogenes* (curve f). The above test results were consistent with the EIS results, providing further evidence for the successful construction of the sensor.

### 3.5. Optimization of Experimental Parameters

The test conditions of the sensor were optimized, and the concentration of AA, the pH value of the electrolyte, the reflow time of the quantum dots, and the incubation time of the electrode with *Listeria monocytogenes* and Exo I were investigated.

AA acts as an electron donor and plays a significant role in increasing photocurrent. As shown in Figure 12a, when there was no AA in the electrolyte, the photocurrent of WO_3_/FTO was the smallest. With the gradual increase in AA concentration, the photocurrent reached its maximum value at 0.10 mol/L. When the AA concentration exceeded 0.10 mol/L, the current value gradually decreased. It may be that the excessive AA concentration increases the absorbance of the electrolyte and reduces the light intensity on the electrode surface. Therefore, 0.10 mol/L was chosen as the optimum concentration of AA.

In addition, aptamers can only remain active in relatively neutral solutions. Figure 12b shows that the WO_3_/FTO electrode photocurrent reached the maximum when the electrolyte pH was 7.4, so 7.4 was selected as the optimal pH.

Three CdTe quantum dots with different reflow times were prepared. It can be seen from the above that the positions of the strongest peaks of the ultraviolet–visible absorption spectra and fluorescence spectra of CdTe quantum dots with different reflow times are different. As time increased, the positions of the strongest peaks moved toward the long-wavelength direction. To further prove the size of the photocurrent of quantum dots at different reflow times, three different QDs were used to construct aptamer electrodes and perform photocurrent detection. Figure 12c shows that the photocurrent was the largest when the reflux time was 3 h. Therefore, CdTe QDs with a reflow time of 3 h were chosen to construct an aptamer sensor.

The incubation time of Exo I and *Listeria monocytogenes* also affects the change in photocurrent. Proper incubation time not only reduces the time used in the preparation process but also enables the aptamer sensor to achieve optimal performance. As can be seen from Figure 12d, the photocurrent gradually decreased with an increase in time before the incubation time of 60 min. After 60 min, the photocurrent decreased, but it gradually stabilized. Therefore, the optimal incubation time of Exo I and *Listeria monocytogenes* was 60 min.

### 3.6. Analytical Performance of Aptamer Sensor

The prepared aptamer sensor relies on the changes in the photocurrent of the electrodes before and after the addition of pathogenic bacteria to quantitatively and qualitatively detect the pathogenic bacteria. The change in sensor photocurrent is directly related to the concentration of pathogenic bacteria. Figure 13a shows the photocurrent change curve of different concentrations of *Listeria monocytogenes*. It shows that as the *Listeria monocytogenes* concentration increased, the photocurrent gradually decreased. This is because of the combination of a large amount of *Listeria monocytogenes* and Ap on the electrode surface, which weakened the sensitization effect of QDs and reduced the photocurrent. Figure 13b shows that from 1.3 × 10 CFU/mL to 1.3 × 10^7^ CFU/mL, the change in photocurrent had a very good linear relationship with the concentration of pathogenic bacteria. The linear equation obtained was ΔI = 9.76logC_LM_ − 4.44 (R^2^ = 0.9980), and the detection limit was 45 CFU/mL. This may be due to the sensitization of quantum dots and the auxiliary amplification effect of Exo I, allowing the aptamer sensor to have a wider detection range and a smaller detection limit.

### 3.7. Selectivity, Stability, and Reproducibility of the PEC Sensing Platform

Specificity is a significant indicator of the sensor. The specificity of the sensor was characterized by the change of photocurrent before and after incubation with different interferents. A quantity of 1.3 × 10^5^ CFU/mL *Listeria monocytogenes* (A) was selected for detection; *Staphylococcus aureus* (B), *Escherichia coli O157:H7* (C), and *Salmonella typhimurium* (D) at the same concentration were selected as interferents, and physiological saline (E) was used as a blank control. As can be seen from Figure 14a, the photocurrent changed significantly before and after the incubation of *Listeria monocytogenes*. After calculation, the change value RSD was 2.94%. The photocurrent changes in the interferents were similar to those in the blank group. This result indicates that the prepared aptamer sensor has satisfactory specificity for *Listeria monocytogenes*.

To investigate the stability of the sensor, the electrode was incubated with 1.3 × 10^5^ CFU/mL *Listeria monocytogenes*, and Exo I was repeatedly switched on and off within 400 s to observe the change in photocurrent to judge the stability of the sensor. The light source switch interval was 20 s, as shown in Figure 14b. It can be seen from the figure that there was almost no change in the photocurrent within 400 s. After the second electrode was placed in a refrigerator at 4 °C for a week, the photocurrent detection was performed again under the same conditions, and the current value dropped by about 5%, which proved that the sensor had good stability.

Five electrodes were constructed under the same conditions, 1.3 × 10^5^ CFU/mL *Listeria monocytogenes* and Exo I were incubated with photoelectric detection, and the reproducibility of the sensor was analyzed. The experimental calculation showed that the RSD was 2.1%, proving that the prepared aptamer sensor has good accuracy and reproducibility.

### 3.8. Analysis of Real Samples

To verify the feasibility of the sensor to detect *Listeria monocytogenes* in actual samples, we added 1 × 10^2^ CFU/mL, 1 × 10^5^ CFU/mL, and 1 × 10^7^ CFU/mL of *Listeria monocytogenes* to chicken samples that did not contain *Listeria monocytogenes* for photocurrent detection. The *Listeria monocytogenes* concentration was obtained and compared with the actual addition to calculate the recovery rate. As shown in Table 1, for three different dilutions of *Listeria monocytogenes*, the prepared aptamer sensor was used for detection, and the recovery rate met the requirements. This shows that the sensor has great potential in practical applications. 

## 4. Conclusions

In the present study, a photoelectrochemical sensor based on WO_3_ was constructed to detect *Listeria monocytogenes*. When pathogenic bacteria were present, the quantum dots fell off the surface of the electrode, causing the photocurrent to decrease. At the same time, combined with the shearing effect of Exo I, the photocurrent was significantly reduced, thereby amplifying the signal changes before and after the addition of pathogenic bacteria and increasing the detection limit. Finally, the performance of the sensing platform was verified through the aspects of specificity, stability, and repeatability. Under the optimal conditions, a range of 1.3 × 10^1^ to 1.3 × 10^7^ CFU/mL was observed, with a detection limit of 45 CFU/mL. This method may have hopeful prospects for the rapid detection of *Listeria monocytogenes*.

## Figures and Tables

**Figure 1 foods-10-02896-f001:**
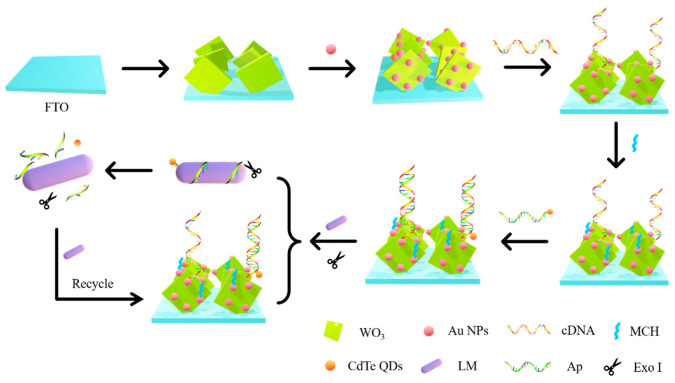
Construction of working electrode and working mechanism of the sensor.

**Figure 2 foods-10-02896-f002:**
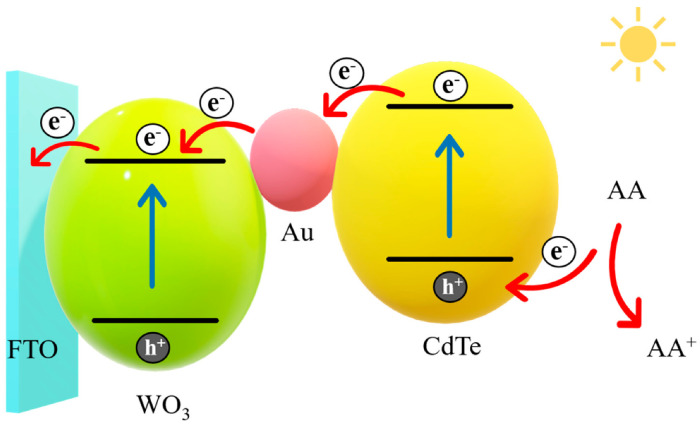
Photogenerated electron transfer mechanism.

**Figure 3 foods-10-02896-f003:**
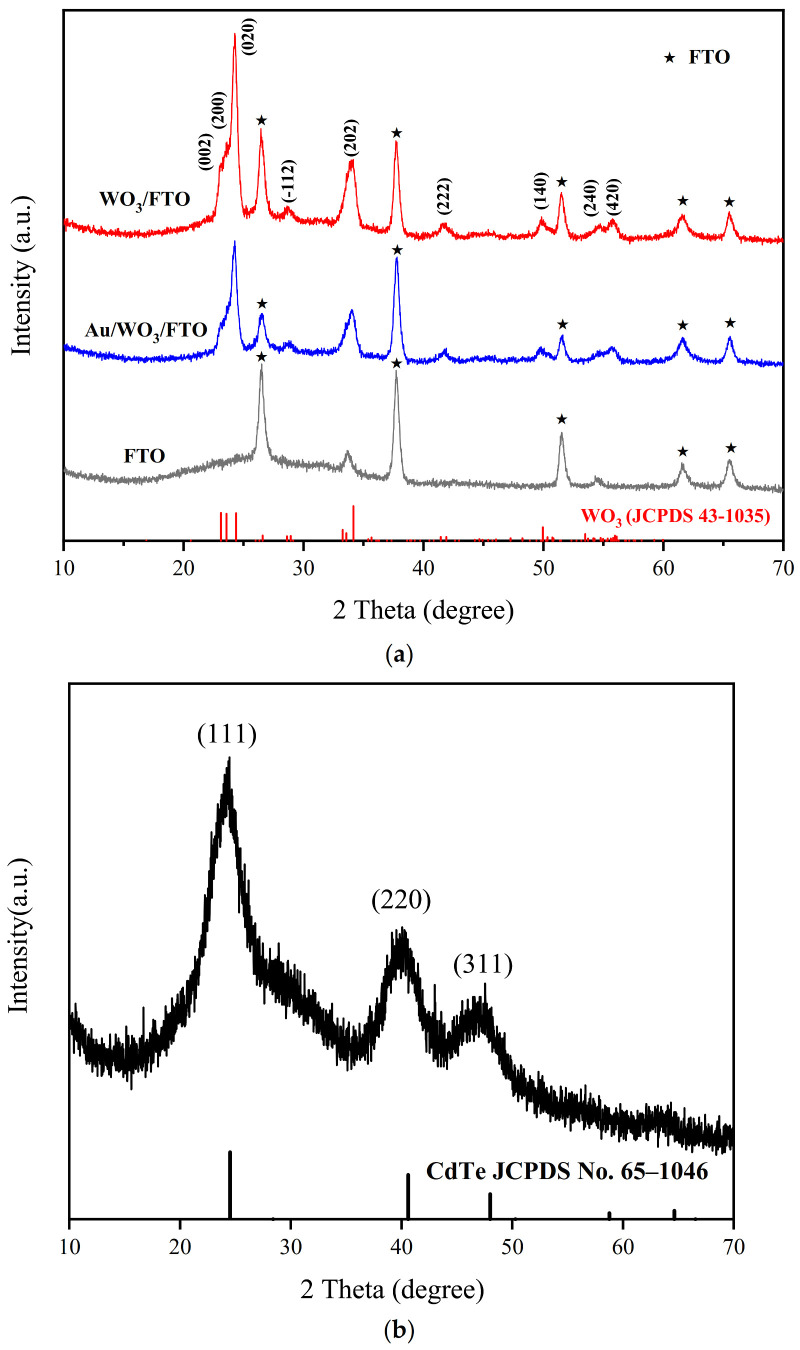
XRD pattern of FTO, WO_3_/FTO, and Au/WO3/FTO (**a**) and XRD pattern of CdTe QDs (**b**).

**Figure 4 foods-10-02896-f004:**
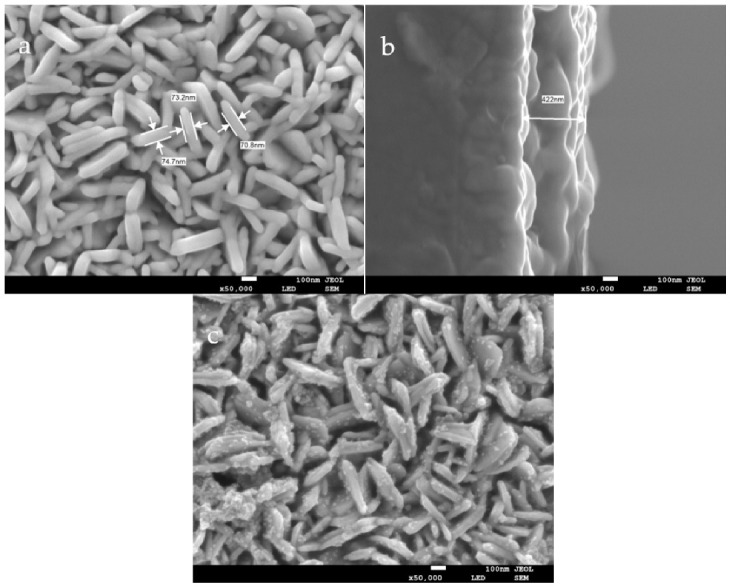
SEM images of the WO3/FTO front (**a**) and cross-section (**b**) and Au/WO3/FTO (**c**).

**Figure 5 foods-10-02896-f005:**
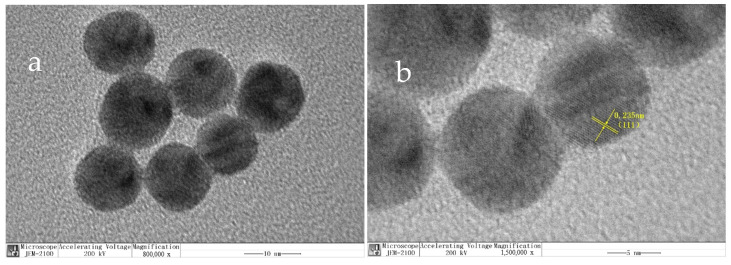
TEM image (**a**) and HRTEM image (**b**) of Au NPs.

**Figure 6 foods-10-02896-f006:**
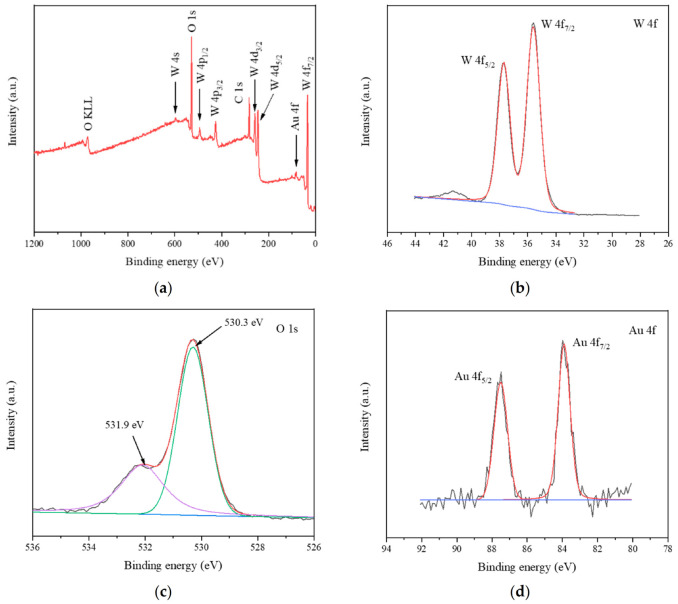
XPS spectra of WO_3_/Au: survey spectrum (**a**), high-resolution XPS of W 4f (**b**), O 1s (**c**), Au 4f (**d**).

**Figure 7 foods-10-02896-f007:**
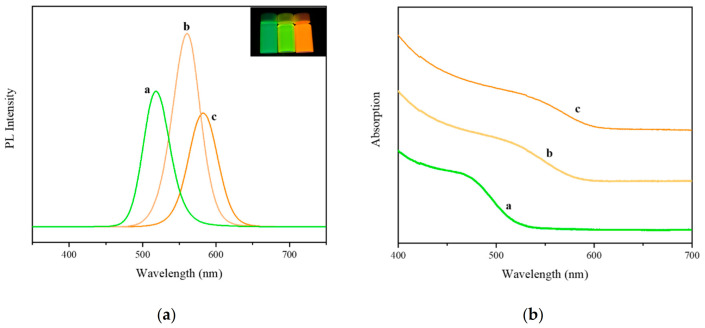
UV-vis (**a**) and fluorescence spectra (**b**) of CdTe QDs, a: 30 min, b: 1 h, c: 3 h. Inset in Figure (**a**) shows the image of CdTe QDs under UV lamps with different reflow times.

**Figure 8 foods-10-02896-f008:**
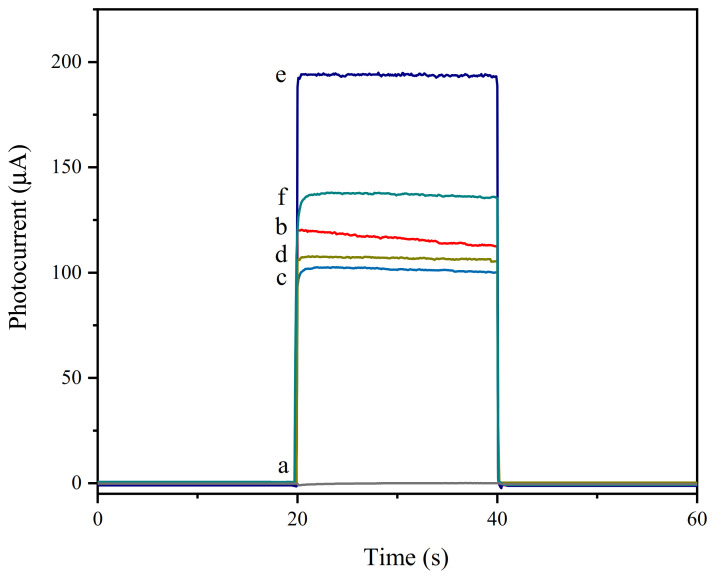
Photocurrent graphs of working electrodes modified with different materials; a: FTO, b: WO_3_/FTO, c: Au/WO_3_/FTO, d: MCH/cDNA/Au/WO_3_/FTO, e: QD–Ap/MCH/cDNA/Au/WO_3_/FTO, f: LM-Exo I/QD–Ap/MCH/cDNA/Au/WO_3_/FTO.

**Figure 9 foods-10-02896-f009:**
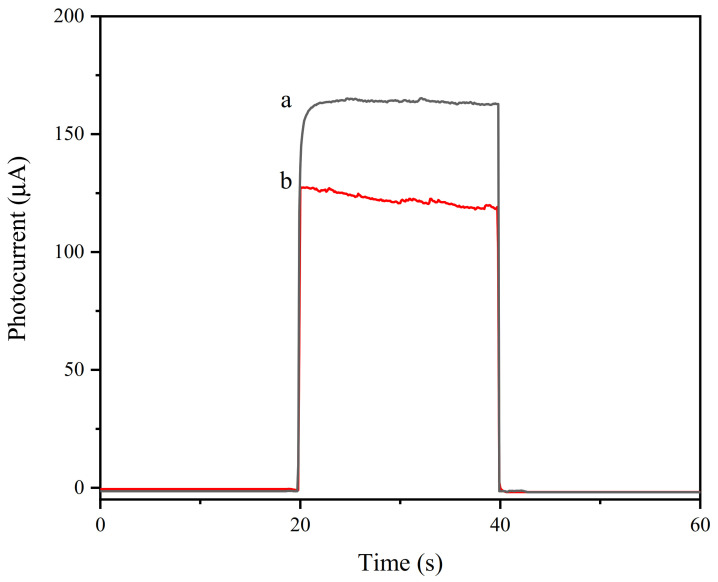
Photocurrent graph of aptamer electrode with or without Exo I modification, a: LM/QD–Ap/MCH/cDNA/Au/WO_3_/FTO, b: LM-Exo I/QD–Ap/MCH/cDNA/Au/WO_3_/FTO.

**Figure 10 foods-10-02896-f010:**
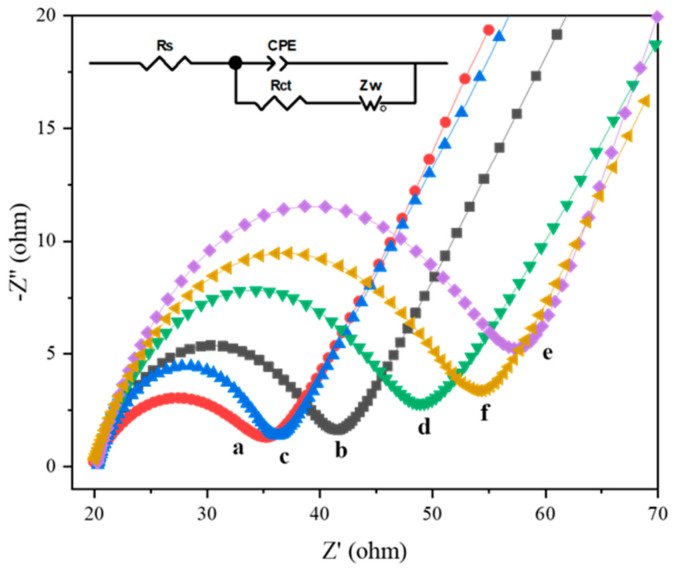
Electrochemical impedance spectroscopy, a: FTO, b: WO_3_/FTO, c: Au/WO_3_/FTO, d: MCH/cDNA/Au/WO_3_/FTO, e: QDs-Ap/MCH/cDNA/Au/WO_3_/FTO, f: LM-Exo I/QDs-Ap/MCH/cDNA/Au/WO_3_/FTO. Inset is the equivalent circuit model.

**Figure 11 foods-10-02896-f011:**
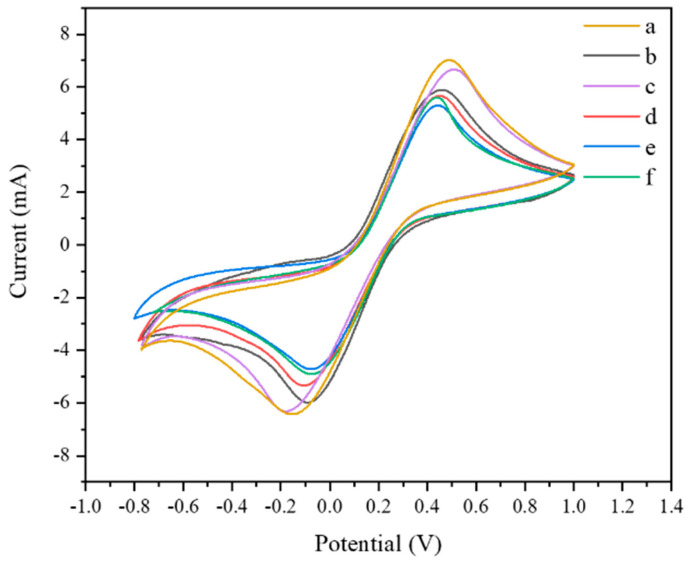
Cyclic voltammogram, a: FTO, b: WO_3_/FTO, c: Au/WO_3_/FTO, d: MCH/cDNA/Au/WO_3_/FTO, e: QD–Ap/MCH/cDNA/Au/WO_3_/FTO, f: LM-Exo I/QD–Ap/MCH/cDNA/Au/WO_3_/FTO.

**Figure 12 foods-10-02896-f012:**
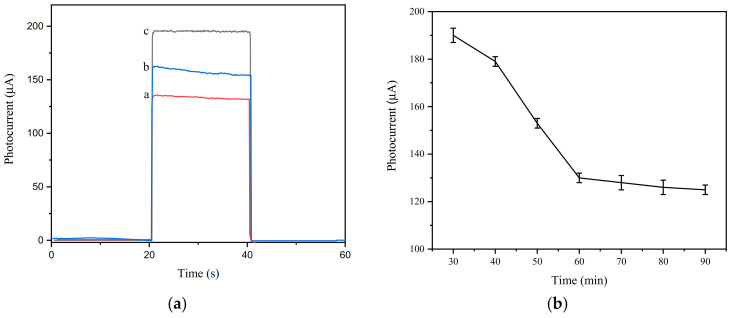
Photocurrent graphs of different AA concentrations (**a**); pH (**b**); quantum dot reflux times (**c**), a: 30 min, b: 1 h, c: 3 h; and incubation time (**d**).

**Figure 13 foods-10-02896-f013:**
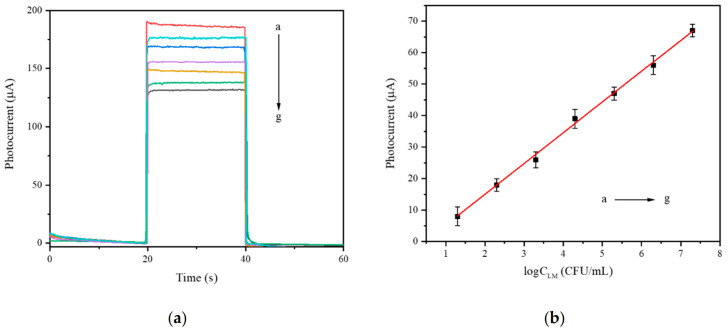
Photocurrent curve (**a**) and a calibration curve (**b**) of the aptamer sensor with different concentrations of *Listeria monocytogenes* (a→g: 1.3 × 10^1^→1.3 × 10^7^CFU/mL).

**Figure 14 foods-10-02896-f014:**
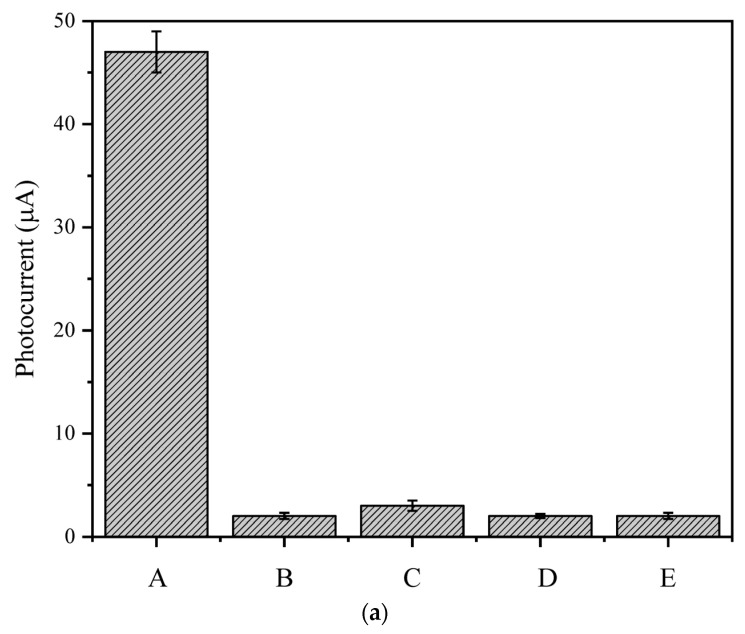
(**a**) Photocurrent response of the sensor to 1.3 × 10^5^ CFU/mL of *Listeria monocytogenes* (A), *Staphylococcus aureus* (B), *Escherichia coli O157:H7* (C), *Salmonella typhimurium* (D), saline (E); (**b**) photocurrent response of the aptamer electrode when the light was turned on and off within 400 s continuously.

**Table 1 foods-10-02896-t001:** Detection of *Listeria monocytogenes* in chicken samples with sensors.

Samples	Added (CFU/mL)	Found (CFU/mL)	Recovery (%)	RSD (%)
1	1 × 10^2^	0.91 × 10^2^	91	2.2
2	1 × 10^5^	1.12 × 10^5^	112	3.4
3	1 × 10^7^	0.89 × 10^7^	89	5.3

## Data Availability

The datasets generated for this study are available on request to the corresponding author.

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
