# Peer review of "A Novel Photoelectrochemical Aptamer Sensor Based on CdTe Quantum Dots Enhancement and Exonuclease I-Assisted Signal Amplification for Listeria monocytogenes Detection"

_foods, 2021, doi:10.3390/foods10122896_

Round 1

Reviewer 1 Report

The paper is very interesting and the description of the sensor technology very accurate.
However, it should be noted that Table 1, relating to the three different dilutions of Listeria monocytogenes, is not present.
I also believe that having performed the test using a single food matrix (chicken) does not allow to have a complete evaluation of the effectiveness of the sensor.
For this purpose it would be useful to test its effectiveness with particularly difficult food matrices such as chocolate or other foods with a high percentage of fat. These factors, in fact, often represent a limit to the effectiveness of the sensors.

Author Response

 According to the reviewer’s suggestion, we showed the Table 1. We understand that using a more complex substrate will be more effective, but the use of chicken for testing has achieved our goal, and Listeria monocytogenes is generally found in under-heated meat and vegetables. The reviewer's comments are very helpful to us.  We will pay special attention in our future work.

Table 1. Detection of Listeria monocytogenes in chicken samples with sensors.

Samples

Added (CFU/mL)

Found (CFU/mL)

Recovery (%)

RSD (%)

1

1×102

0.91×102

91

2.2

2

1×105

1.12×105

112

3.4

3

1×107

0.89×107

89

5.3

Reviewer 2 Report

The authors proposed an interesting research on a novel photoelectrochemical aptamer sensor for L. monocytogenes detection. Materials, methods and results are described clearly and comprehensively. However, it is unclear whether the authors performed an incubation of chicken homogenate after contamination (line184); it should be specified.
Can the authors also explain why they chose chicken meat as a matrix to be tested?
It would also be interesting to know if the protocol was applied to naturally contaminated matrices.

Author Response

We appreciate the reviewer’s suggestion. We did not hatch the chicken homogenate after contamination. We are sorry for the unclear description, and the description (Line 185) has been revised accordingly. Listeria monocytogenes is widely distributed and easily exists in under-heated meat, chicken meat is easy to carry bacteria and it is easy to contaminate the bacteria during the production process. Choosing chicken as the detection substrate can simulate the actual situation and has practical significance. For the naturally polluted substrates, we will continue to study in the future work.

Reviewer 3 Report

The main concern from this reviewer is the specificity of the detection system authors applied, considering the complexity of the actual samples. In other words, this platform lacks "control"(e.g., PCR) for indicating the stability and discrimination of other bacteria or microbes when mixed together (e.g., Fig. 14). Some additional tests should be done to show the performance of the sensing platform. Other issues are the readability and conclusions, which should be toned down based on the results.

Author Response

Thank the reviewers for these precious comments and suggestions. We understand that some additional tests may better reveal the stability and discrimination. However, in the present study, we think that this test can simply explain the specificity of the sensor. At the same time, we think your points are excellent, and we will continue the study in the future work.

In the conclusion, we have also made relevant changes for the readability as follows:

The photocurrent response is improved by quantum dots. Depending on the shear action of exonuclease, more quantum dots are moved away from the electrode surface through circulation, which also significantly enhanced the photocurrent response. Under the optimal conditions, a range of 1.3×101 to 1.3×107 CFU/mL with a detection limit of 45 CFU/mL was determined.

Reviewer 4 Report

The submitted manuscript is of a very high quality and presents results of the comprehensive study focused on the design, synthesis and analysis of the new method of detection of Listeria monocytogenes in food samples. The level of novelty is high, experiments (both physicochemical and microbiological) have been designed and performed properly. The discussion is rich and supported by the results. I only wish that the Author describe one more aspect in the discussion that is the cost of their method in comparison to the standard ones. Also, the level of English used in the introduction section should be increased. Apart from that I have some detail comments that are presented below.

Line 19, it should be 1.3 x 101 – 1.3 x 107. In the present way it is very confusing.

Lines 25-26, “cause” is used three times in a row, please rewrite.

Line 36, what do you mean by “detection stability”?

Line 42, it should be “professionals”.

Line 45, it should be “they”, not “it”.

Line 64-65, short information about the QDs should be placed here.

Line 104, more information about PXRD measurements should be described here, i.e. temperature, radiation source, wavelength, 2-theta range, resolution, etc.

Line 108, spectra are “recorded”, not “obtained”

Line 118, ethanol is not listed in section 2.1.

In many places, i.e. lines 27, 28, 31, Listeria monocytogenes is not written using italics.

Line 181, the abbreviations used in the Figures must be explained here.

Line 183, “chicken” sample is not listed in the section 2 (experimental)

Author Response

 We appreciate the reviewer’s suggestion. According to the editor’s suggestion, the relative aspects have been adjusted in the manuscript.

Line 19, 13-1.3×107 CFU/mL have been revised as 1.3×101-1.3×107 CFU/mL.

Lines 25-26,  “cause”  have been revised as “can lead to” and “have aroused”.

Line 36,  it has been revised as “the detection accuracy is high”.

Line 42, “professional” has been revised as “professionals”.

Line 45, “it” has been revised as “they”

In line 64-65, “QDs are a kind of nanoscale semiconductor, such as CdS, CdSe, CdTe.”have been added here.

Line 104, “The test conditions were set to be Cu-Ka radiation source, scan rate of 5°/min, and scan range of 10–70 degrees at room temperature.”have been added here.

Line 108, “obtained”has been revised as “recorded”.

Line 118, “ethanol”has been added in section 2.1.

lines 27, 28, 31 i.e., Listeria monocytogenes has been revised using italics.

Line 181, 6-mercapto-1-hexanol (MCH), CdTe Quantum dots-aptamers (QDs-Ap), and Exonuclease I (Exo I) have been added in 2.8.

Line 183, “Raw chicken purchased from local supermarket.” have been added in 2.1

Round 2

Reviewer 1 Report

I confirm the importance and value of this research but I remain of the opinion that to make the results obtained significant it is necessary to test and compare the efficacy with other food matrices, much more complex than poultry meat.

Author Response

Answer: We appreciate the reviewer’s suggestion. In fact, we intend to test and compare the efficacy with other food matrices. Unfortunately, due to the epidemic caused by COVID-19, we are not able to start work immediately. We may be not able to carry out the supplementary experiments in recent.

Reviewer 3 Report

The authors did not revise the manuscript according to reviewers' comments. Extensive modifications are required before acceptance.

Author Response

The main concern from this reviewer is the specificity of the detection system authors applied, considering the complexity of the actual samples. In other words, this platform lacks "control"(e.g., PCR) for indicating the stability and discrimination of other bacteria or microbes when mixed together (e.g., Fig. 14).

Answer: We appreciate the reviewer’s suggestion. We used different pathogenic bacteria to simply detect the specificity of the sensing platform. We understand that this method of detecting specificity is flawed.  Unfortunately, due to the epidemic caused by COVID-19, we are not able to start work immediately. We plan to combine with existing detection methods (PCR and other technologies) as a verification comparison in the future. We will devote more energy to research this aspect of work in the future.

Some additional tests should be done to show the performance of the sensing platform.

Answer: According to the literatures, the performance characterization of the sensing platform, there are only three tests for specificity, stability and repeatability. How to characterize the performance of the sensing platform more comprehensively, we will further add in future work. Now due to the epidemic caused by COVID-19, it is inconvenient to carry out work.

 Other issues are the readability and conclusions, which should be toned down based on the results.

Answer: According to the reviewer’s suggestion, we have revised the language of this manuscript improve its readability, and moreover, some conclusions were toned down based on the results.

Lines 25-26,  “cause”  have been revised as “can lead to” and “have aroused”.

Line 36,  it has been revised as “the detection accuracy is high”.

Line 42, “professional” has been revised as “professionals”.

Line 45, “it” has been revised as “they”

In line 64-65, “QDs are a kind of nanoscale semiconductor, such as CdS, CdSe, CdTe.”have been added here.

Line 104, “The test conditions were set to be Cu-Ka radiation source, scan rate of 5°/min, and scan range of 10–70 degrees at room temperature.”have been added here.

Line 108, “obtained”has been revised as “recorded”.

Line 118, “ethanol”has been added in section 2.1.

lines 27, 28, 31 i.e., Listeria monocytogenes has been revised using italics.

Line 181, 6-mercapto-1-hexanol (MCH), CdTe Quantum dots-aptamers (QDs-Ap), and Exonuclease I (Exo I) have been added in 2.8.

Line 183, “Raw chicken purchased from local supermarket.” have been added in 2.1.

Line 419: “outstanding specificity” was changed as “satisfactory specificity for Listeria monocytogenes.”

Line 18: “pathogenic bacteria” was changed as “Listeria monocytogenes”  

The conclusion section was revised as follows:

In the present study, a photoelectrochemical sensor based on WO3 was constructed to detect Listeria monocytogenes. When pathogenic bacteria are present, the quantum dots will fall off the surface of the electrode, causing the photocurrent to decrease. At the same time, combined with the shearing effect of Exo â… , the photocurrent is significantly reduced, thereby amplifying the signal changes before and after the combination of pathogenic bacteria, and increasing the detection limit. Finally, the performance of the sensing platform is verified from the aspects of specificity, stability and repeatability. Under the optimal conditions, a range of 1.3×101 to 1.3×107 CFU/mL with a detection limit of 45 CFU/mL. This method would have hopeful prospects in rapid detection of Listeria monocytogenes.

Reviewer 4 Report

The Authors have significantly revised their manuscript. 

Author Response

Answer: We appreciate the reviewer’s suggestion.